

# Mitigating NaCl stress in *Vigna radiata* L. cultivars using *Bacillus pseudomycoides*

Bushra Bilal[1], Zafar Siddiq[1], Tehreema Iftikhar[1] and
Muhammad Umar Hayyat[2]

[1] Department of Botany, Government College University Lahore, Lahore, Punjab, Pakistan
[2] Sustainable Development Study Center, Government College University Lahore, Lahore, Punjab, Pakistan

## ABSTRACT

Salt stress is one of the significant abiotic stress factors that exert harmful effects on plant growth and yield. In this study, five cultivars of mung bean (*Vigna radiata* L.) were treated with different concentrations of NaCl and also inoculated with a salt-tolerant bacterial strain to assess their growth and yield. The bacterial strain was isolated from the saline soil of Sahiwal District, Punjab, Pakistan and identified as *Bacillus pseudomycoides*. Plant growth was monitored at 15-days interval and finally harvested after 120 days at seed set. Both sodium and potassium uptake in above and below-ground parts were assessed using a flame photometer. Fresh and dry mass, number of pods, seeds per plant, weight of seeds per plant and weight of 100 seeds reduced significantly as the concentration of NaCl increased from 3 to 15 dSm$^{-1}$. There was a significant reduction in the growth and yield of plants exposed to NaCl stress without bacterial inoculum compared to the plants with bacterial inoculum. The latter plants showed a significant increase in the studied parameters. It was found that the cultivar Inqelab mung showed the least reduction in growth and yield traits among the studied cultivars, while Ramzan mung showed the maximum reduction. Among all the cultivars, maximum Na$^+$ uptake occurred in roots, while the least uptake was observed in seeds. The study concludes that NaCl stress significantly reduces the growth and yield of mung bean cultivars, but *Bacillus pseudomycoides* inoculum alleviates salt stress. These findings will be helpful to cultivate the selected cultivars in soils with varying concentrations of NaCl.

## INTRODUCTION

Salinity imposes enormous pressure on agricultural production due to the deposition of salt in soil (*Shahrasbi et al., 2020*). Salt stress has adversely affected about 20% of cultivated land and 50% of cropland worldwide (*Kamran et al., 2020*). Anthropogenic activities and climate change accelerate the expansion of salt-affected area (*Arora & Bhatla, 2017*). Salinity reduces about 50% of agricultural productivity in arid and semiarid regions (*Flowers, Galal & Bromham, 2010*). In Pakistan, about 6.7 million hectares of land is affected by salinity (*Hassan et al., 2021*). Salinity degrades the soils of salt-affected sites due to improper salt management (*Saleem et al., 2020*). Soluble salts are retained in soil during agricultural practices due to excessive use of nitrogenous fertilizers, improper irrigational

Corresponding author
Bushra Bilal,
bushrabilal240@gmail.com

practices, poor drainage, high evaporation rates, low rainfall, and irrigation with salt-containing water, all of which contribute to soil salinity (*Zafar et al., 2019*; *Okur & Orçen, 2020*). At present, there is increase in both drought and salinity, resulting in the decline of crop productivity. This needs to be addressed by screening different cultivars of crops against these stress factors (*Ma, Dias & Freitas, 2020*; *Oyebamiji et al., 2024*; *Huanhe et al., 2024*). Salinity-induced stress causes numerous biochemicals and physiological damages in crops, including imbalanced ion up-take, and production of reactive oxygen species (ROS). Accumulated ROS ultimately results in oxidative stress, causing damage to, plant tissues, breakage of nucleic acids and shrinkage of cell, which harms cellular structure (*Rehman et al., 2019a*; *Debnath et al., 2021*). More concentration of reactive oxygen species disrupts enzymatic processes, cell membranes, as well as degradation of lipids, proteins, and other biochemicals (*Shahid et al., 2020*; *Xu et al., 2022*). High concentrations of $Na^+$ lead to higher $Na^+/Ca^+$ and $Na^+/K^+$ ratio, which consequently results in less absorption of $K^+$ and $Ca^+$ that disrupts cellular membranes and enzymatic activities, causes imbalances in cellular equilibrium and destroys cellular balance (*Kamran et al., 2019*). An increased amount of salt in the soil profile also results in tissues dryness through less absorption of water, ion toxicity, and reduced osmotic potential (*Safdar et al., 2019*; *Garcia-Capparos et al., 2021*). The difference in water-availability and nutrient uptake disturbs and ceases the germination of seeds and growth of plants (*Nawaz et al., 2021*). The responses of crops to salt stress as well as their developmental stages vary across crops and their selected cultivars (*Alsafran et al., 2022*). The use of bio-inoculants, such as growth promoting bacteria has been proved to be highly useful in enhancing crop production against different salt stresses (*Kumawat, Nagpal & Sharma, 2022*; *Hyder et al., 2023*). These bacteria can reduce the harmful effects of abiotic stress factors (*Chieb & Gachomo, 2023*). Soil is composed of diverse microbes, including fungi, algae and bacteria, as well as minerals, which interact with the plants' roots (*Chen et al., 2023*). In rhizosphere, soil bacteria help to prevent diseases and enhance the smooth uptake of nutrients (*Hanelt & Muller, 2013*; *Prittesh et al., 2020*; *Kong et al., 2021*). The induction of bio-formulation with plant growth-promoting bacteria aids in crop long-term fertility and makes modern approaches more appealing than traditional approaches (*Backer et al., 2018*). The rhizosphere is the area in the soil surrounding plant roots where soil microbes are present in abundance and perform microbial activities (*Grover et al., 2021*). Rhizo-microbial activity depends on soil properties, climate changes, and host species (*Lyu et al., 2020*). Bacterial inoculation enhances plant growth through direct and indirect actions (*Saleemi et al., 2017*). Direct mechanisms involve the easy uptake of nutrients from natural resources or phytohormone secretions that improve plant growth (*Basu et al., 2021*). Indirect mechanisms include enhancing stress tolerance as well as suppressing plant pathogens (*Fatima & Arora, 2021*). The diversity of soil and microbes is the backbone of agroecosystem and can increase crop yield even under stress (*Fatima et al., 2020*). Beneficial microorganisms dwelling in the soil have positive influence on plant metabolism, allowing plants to survive under abiotic stresses, like salt, drought, heat and heavy metal stress (*Luna et al., 2007*; *Numan et al., 2018*). Among others, few bacterial species, such as Bacillus, Enterobacter, Azospirillum, Klebsiella, Pseudomonas, Arthobacter, and Serratia enhance different crops growth and

nutrient uptake efficiency supporting the production of phytohormones (indole acetic acid and cytokinin) (*Rosenberg, Gutnick & Rosenberg, 1980*; *Khoso et al., 2024*). Apart from this, cytokinins are also involved in the management of environmental stresses by transferring signaling from roots to shoots (*Ansari, Ahmad & Pichtel, 2019*). Different bacterial strains, such as *Bacillus pseudomycoides*, *Bacillus massilioanrexius*, *Bacillus thuringiensis*, *Serratia marcescens*, and *Acinetobacter* sp. have different defense mechanisms against stress (*Saha, Chatterjee & Biswas, 2010*). *Bacillus pseudomycoides* is a facultative anaerobe, spore-forming, mesophilic bacterium that forms rhizoid colonies (*Paul et al., 2022a*). It can produce different volatile compounds that result in increased biomass in plant roots, disease resistance, abiotic stress tolerance, plant endogenous auxin homeostasis, and ion uptake (*Farag, Zhang & Ryu, 2013*; *Zhang et al., 2018a*; *Gupta et al., 2021*). However, the use of these microbes has to be tested across different crops, such as wheat, alfalfa, sweet corn under various stress conditions (drought and salinity stress *etc.*,) (*El-Saadony et al., 2019*; *Tamindžija et al., 2019*; *Paul et al., 2022b*; *Knezevic et al., 2021*; *Katsenios et al., 2022*).

Mung bean (*Vigna radiata* L.) is among the benefcent diet, having ecological and economic values. It is a summer crop, and it is mostly cultivated in arid and semi-arid regions (*Shafique et al., 2023*). The main dietary components of mung bean are protein, carbohydrates, fiber, fatty acids, and amino acids (*Aasim et al., 2019*). Due to the nitrogen-fixing ability of mung beans, they can be used for soil reclamation and enhancement of soil fertility (*Nigar et al., 2023*). Globally, the annual production of this crop is 30 million tons, which is 5.1% of the total pulses produced (*Sehrawat et al., 2018*). The major countries that cultivate mung bean include: Africa, Burma, China, India, Pakistan, and Queensland (*Pandey et al., 2021*). Mung bean is considered a salt-sensitive crop and, therefore, it is susceptible to salt stress (*Kumar et al., 2023*). About 70% of mung bean production was reduced at 50 mM NaCl concentration (*Dutta & Bera, 2014*). There are a number of mung bean cultivars that have been investigated against different stress conditions; *e.g.*, Chackwal-Mung-06, NCM-2013, AZRI-06, NM-11, (*Ullah et al., 2016*), Pant M-8, KM-2241, IPM 02-3, (*Katiyar et al., 2019*), Mung MgAT-7, MgAT-4, NCM-13, (*Iqbal et al., 2022*), MMP-15001-19, PRI-2018, (*Khalid et al., 2021*; *Ali et al., 2022*). The five mung bean cultivars (Inqelab mung, NIFA-19, NIFA-17, Sona mung, Ramzan mung) selected in this study have not been studied against salt stress. The main objectives of the current study are to: (1) investigate the effect of NaCl stress with and without bacterial inoculum on growth, yield, and physiological parameters of different cultivars of *Vigna radiata* L. and their comparison, and (2) assess the uptake of $Na^+$ and $K^+$ in root, stem, leaves and seeds among the five selected cultivars.

## MATERIALS AND METHODS

### Bacterial isolation, screening, and characterization

Three different saline areas in Jahan Khan, Sahiwal District, Punjab, Pakistan were chosen for soil sampling, with geographical location of 30°34′0″N, 73°11′0″E. Salt-tolerant bacteria were isolated from salt-affected soils (*Manasi, Rajesh & Rajesh, 2016*). Screening of salt-tolerant bacteria was carried out following *Hena et al. (2022)*. Sample of highly salt-tolerant bacterial strain was prepared and sent to Humanizing Genomics Macrogen,

Korea for molecular characterization and identification, based on the method of *Sanjay et al. (2018)*.

## Preparation of bacterial inoculum

Bacterial inoculum was prepared by mixing 8 g of nutrient broth (NB) into one liter of distilled water and one loop of the respective bacterial strain was inoculated in it. It was placed in an incubator at 37 °C for 24 h to prepare slants for inoculum application.

## Seeds collection

Seeds of five different cultivars *i.e.*, NIFA-19, NIFA-17, Ramzan mung, Inqelab mung and Sona mung were collected from the Nuclear Institute for Food and Agriculture Peshawar and the Agriculture Research Institute Dera-Ismael Khan, Pakistan (*Khattak et al., 2006, 2020*; *Mansoor et al., 2017*; *Khattak, Saeed & Ibrar, 2019*).

## Experimental set-up

The pot experiment was conducted to measure plant growth, yield, and physiological parameters of the five selected cultivars of *V. radiata* (Inqelab mung (V1), NIFA-19 (V2), NIFA-17 (V3), Sona mung (V4) and Ramzan mung (V5)). A total of twelve treatments were applied in a completely randomized design (CRD) for each cultivar of mung bean. The experimental design was in the following pattern: (i) plants without NaCl application and *B. pseudomycoides* inoculation ($T_0$) (ii) plants without NaCl application but with *B. pseudomycoides* inoculation ($T_0B$) (iii) plants with 3 dSm$^{-1}$ NaCl ($T_1$) (iv) plants with 3 dSm$^{-1}$ NaCl and *B. pseudomycoides* ($T_1B$) (v) plants with 6 dSm$^{-1}$ NaCl ($T_2$) (vi) plants with 6 dSm$^{-1}$ NaCl and *B. pseudomycoides* ($T_2B$) (vii) plants with 9 dSm$^{-1}$ NaCl ($T_3$) (viii) plants with 9 dSm$^{-1}$ NaCl and *B. pseudomycoides* ($T_3B$) (ix) plants with 12 dSm$^{-1}$ NaCl ($T_4$) (x) plants with 12 dSm$^{-1}$ NaCl and *B. pseudomycoides* ($T_4B$) (xi) plants with 15 dSm$^{-1}$ NaCl ($T_5$) (xii) plants with 15 dSm$^{-1}$ NaCl and *B. pseudomycoides* ($T_5B$). The seeds of the cultivars were germinated in the Botanic Garden and five days old seedlings were treated with 3 to 15 dSm$^{-1}$ concentration of NaCl. They were compared with the control without NaCl. The bacterial inoculum was applied in the pots having germinated seeds. Each pot had four replicate plants, and each treatment had four pots (*Gamalero & Glick, 2022*). The pots were filled with soil from Botanic Garden. Physico-chemical properties of Botanic Garden soil were obtained from *Rizwan, Hayyat & Farooq (2013)*. 10 mL of nutrient broth having bacterial inoculum was mixed with distilled water and poured into the pots. The experiment was carried out for 120 days (March to June 2023) under ambient conditions at the Botanic Garden of Government College University Lahore Pakistan. The experimental conditions included a mean day-time temperature of 38 °C, relative humidity of 26% and solar radiation of 802 watts/m$^2$. The plants were monitored on weekly basis for their growth and irrigated appropriately to avoid any water stress. Equal amount of water was applied to all pots.

## Mid and final harvesting

During mid-harvest, fresh and dry weight of plant parts (above and below ground) was measured. At final harvest, fresh and dry weight of plant parts along with other yield

attributes *i.e.*, number of pods per plant, total seed number per plant, total seed weight per plant and weight of 100 seeds were measured (*Desai et al., 2022*).

## Physiological measurements

The physiological parameters, including maximum photosynthetic rate, stomatal conductance, and transpiration rate were measured using IRGA LCA4. These measurements were taken under ambient conditions on a clear-sunny day from 10:30 am to 11:30 am, just before the mid-harvest of plants. The chlorophyll contents were measured using a spectrophotometer (Spectroscan 80D) (*Yoshida, Forno & Cock, 1971*). The intrinsic water use efficiency ($\mu$mol $CO_2$ $mol^{-1}$ $H_2O$) was calculated based on *Ferreira et al. (2018)*.

## Na$^+$ and K$^+$ uptake

In order to assess the Na$^+$ and K$^+$ uptake, the ash of leaves, stem, seeds, and roots of the studied cultivars was prepared by heating the samples in muffle furnace at 550 °C for 3 h (Gallenhamp Size 2). The prepared ash was dissolved in 5 mL of 2N HCl, and allowed to stand for 15 min, after which 50 mL of distilled water was added. The solution was filtered by using Whattman No. 42 filter paper to have an extract of a particular plant part. The uptake of sodium and potassium was assessed using flame photometer (S20 Spectro lab). Data were collected as previously described by *Rahman et al. (2016)*.

## Data analysis

Data were statistically analyzed using one way ANOVA (Duncan's Multiple Range test (DMRT)). It involved comparing the means across the treatments using the Statistix v. 8.1. Sigma Plot 14.0 version was used to analyze the descriptive statistics, and correlations between parameters as well as creation of graphics.

# RESULTS

## Bacterial screening, identification, and characterization

Thirteen bacterial isolates were obtained from salt contaminated soil of Sahiwal. Nine bacterial isolates exhibited tolerance at 10,000 ppm; three isolates tolerated stress at 20,000 ppm. Only one isolate showed growth up to 30,000 ppm of NaCl. After screening, only one bacterial strain was chosen for inoculation due to its higher tolerance against NaCl. This is because the bacterial cell population significantly decreased from 10.23 to 4.7 CFU/mL in two strains while selected strain showed less reduction from 10.56 to 7.23 CFU/mL as NaCl concentration increased. The pure isolate was *Bacillus pseudomycoides*, which is a gram-positive, non-motile, long rod-shaped, and spore-forming bacterium under light microscope. Figure 1 shows the homology of the isolated bacterial strain.

## Effect of NaCl on *V. radiata* cultivars biomass

A significant difference in growth and yield among all the cultivars was observed across all the cultivars treated with NaCl; while the plants treated with *Bacillus pseudomycoides* inoculum withstood the drastic effect of NaCl (Fig. 2). During the mid-harvest, across the different cultivars treated with *Bacillus pseudomycoides* inoculum (from Inqelab mung to

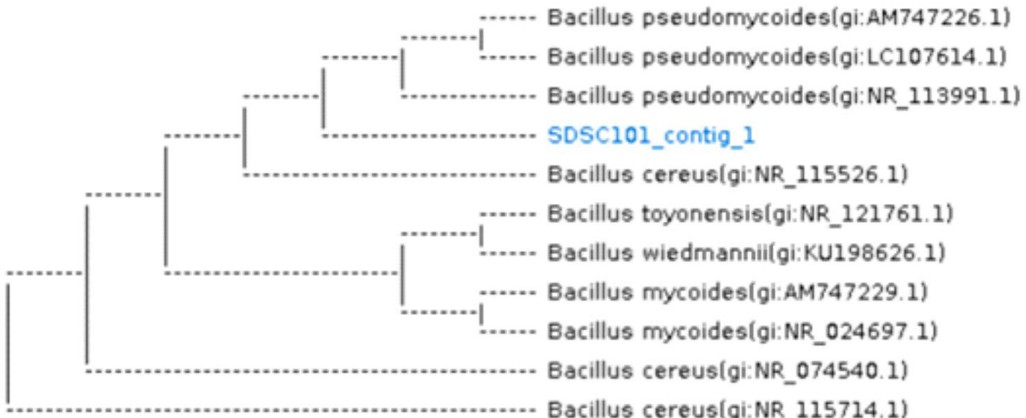

**Figure 1 Dendrogram showing the homology of *Bacillus pseudomycoides* bacterium (16S rRNA report).** SDSC101_contig_1 showing the present study bacterial sample.

Ramzan mung), plant fresh weight showed a significant increase from 63.8–93.8%, while it significant decreased from 57.5–13.6% in those treated with 3–15 dSm$^{-1}$ NaCl treatments. Similarly, across the cultivars treated with *Bacillus pseudomycoides* inoculum and NaCl treatment (from Inqelab mung to Ramzan mung), plant dry weight showed a significant increase from 72.3–84.8%; while it decreased from 38.6–16.4% in those treated with 3–15 dSm$^{-1}$ NaCl treatment (Fig. S1).

At final harvest, a significant increase from 71.8–94.3% in fresh weight of plant was observed across the cultivars treated with *Bacillus pseudomycoides* inoculum (from Inqelab mung to Ramzan mung); while it had a significant decrease from 57.3–14.5% in those treated with salt treatments (3–15 dSm$^{-1}$) (Fig. S2, Supplemental Material). Among all the cultivars treated with *Bacillus pseudomycoides* inoculum and salt stress (3–15 dSm$^{-1}$), plant dry weight also showed significant increase from 67.3–92.2% while it decreased from 45.9–14.7% (Fig. 2). All five studied cultivars of *V. radiata* (Inqelab mung, NIFA-19 mung, NIFA-17 mung, and Sona mung and Ramzan mung) showed significant differences in dry weight (above and below ground) (Figs. 2A–2E). The measured differences in the fresh and dry weight of cultivars were; V5 < V4 < V3 < V2 < V1.

## Effect of NaCl on yield

The yield attributes, such as number of nodules, number of pods per plant, number of seeds per pod per plant, weight of seeds per pod per plant, and weight of 100 seeds also showed reduction in plants without bacterial inoculum, from $T_0$-$T_5$ in all the cultivars. The plants inoculated with *Bacillus pseudomycoides* had more yield than plants treated with salt only. There were significant differences observed in yield parameters among the *Vigna radiata* cultivars, Inqelab mung (Figs. 3A, 3F, 3K, 3P and 3U) NIFA-19 mung (Figs. 3B, 3G, 3I, 3Q, 3V), NIFA-17 mung (Figs. 3C, 3H, 3M, 3R and 3W), Sona mung (Figs. 3D, 3I, 3N, 3S and 3X), and Ramzan mung (Figs. 3E, 3J, 3O, 3T and 3Y). Across the five cultivars treated with NaCl treatment and *Bacillus pseudomycoides* inoculum, percentage increase in number of root nodules was 74.3–87.5%; pods, 65.9–92.8%; seeds, 72.8–94.5%;

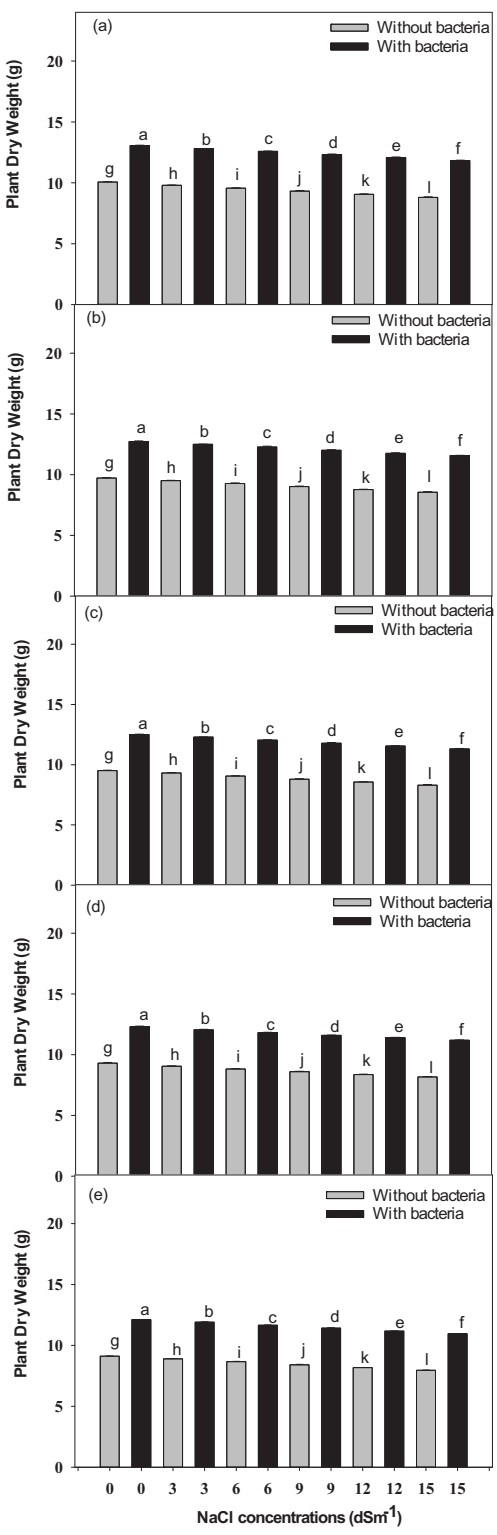

**Figure 2 The mean dry weight of the studied plants-Inqelab mung (A), NIFA-19 (B), NIFA-17 (C), Sona mung (D) and Ramzan mung (E).** Bars indicate mean (±) standard error and different letters represent significant differences. Grey bars indicate soil with NaCl treatment, while black bars indicate NaCl treatment combined with *B. pseudomycoides* inoculum.

weight of seeds, 64.7–85.9% and weight of 100 seeds, 74.9–94.8%. Similarly, in the plants (from Inqelab mung to Ramzan mung respectively) treated with NaCl treatment only, percentage decrease in number of root nodules ranged from 32.8–12.7%; pods, 27.9–9.6%; seeds, 33.7–15.7%; weight of seeds, 45.8–15.6% and weight of 100 seeds, 25.7–7.9%.

## Effect of NaCl on Gaseous exchange and $Na^+$, $K^+$ uptake

Significant differences were found among the studied cultivars in terms of their maximum photosynthetic rate transpiration, stomatal conductance and water use efficiency (Figs. 4A–4T). Plants (Inqelab mung to Ramzan mungy) inoculated with *Bacillus pseudomycoides* showed percentage increase in the following order: photosynthetic rate (75.3–83.8%), transpiration rate (53.85–87.1%) and stomatal conductance (67.7–81.8%); while they showed percentage decrease in the following order under different salt treatments (3–15 $dSm^{-1}$): photosynthetic rate (25.8–9.4%), transpiration rate (36.2–17.3%) and stomatal conductance (47.4–13.5%). Among the cultivars (from Inqelab mung to Ramzan mung) treated with only NaCl treatment (3–15 $dSm^{-1}$), water use efficiency also showed increase from 68.3–92.1%, while when the cultivars were treated with *Bacillus pseudomycoides* inoculum, it decreased from 28.3–13.8%. The up-take of $Na^+$ and $K^+$ varied in both the above and below ground parts of all varieties. The $Na^+$ concentration was higher in roots than in stems and leaves, while the minimum amount of $Na^+$ was detected in seeds. The amount of $Na^+$ in mung bean plants belonging to all the cultivars treated with salt treatments and with and without *B. pseudomycoides* ranged from 0.67–2.77 g $kg^{-1}$ in roots (Figs. 5A–5E), 0.40–2.83 g $kg^{-1}$ in stems (Figs. 5F–5J), 0.13–2.26 g $kg^{-1}$ in leaves (Figs. 5K–5O) and 0.09–1.90 g $kg^{-1}$ in seeds (Figs. 5P–5T) Similarly, the concentration of $K^+$ ranged from 6.73–11.56 g $kg^{-1}$ in roots (Figs. 6A–6E), 7.25–10.83 g $kg^{-1}$ in stems (Figs. 6F–6J), 7.20–10.40 g $kg^{-1}$ in the leaves (Figs. 6K–6O) and 7.03–8.76 g $kg^{-1}$ in the seeds (Figs. 6P–6T), across the cultivars treated with salt treatments only (3–15 $dSm^{-1}$).

Significant difference in the $Na^+/K^+$ ratio was also found in the studied cultivars (Fig. S3, Supplemental Material). The increase in ratio found across the cultivars inoculated with *Bacillus pseudomycoides* was in the following order: 63.2–91.2% in roots, 72.7–88.3% in stem, 73.8–87.6% in leaves, 69.3–91.6% in seeds, while decrease was found in those treated with 3–15 $dSm^{-1}$ NaCl treatment in the following order: 38.4–16.7% in roots, 43.8–17.7% in stem, 34.8–12.7% in leaves, 28.4–11.7% in seeds.

## Effect of NaCl on Chlorophyll contents

Figures 7K–7O shows the total chlorophyll contents across the studied cultivars. Chlorophyll a ranged from 0.84–2.53 mg $g^{-1}$ (Figs. 7A–7E); chlorophyll b, from 0.39–2.72 mg $g^{-1}$ (Figs. 7F–7J) and total chlorophyll contents, from 1.23–3.95 mg $g^{-1}$ (Figs. 7K–7O). There was a significant increase in chlorophyll a (68.4–87.5%) and chlorophyll b (65.8–84.7%) contents with *Bacillus psedomycoides* inoculum, while there was a decrease in chlorophyll a (31.8–13.7%) and chlorophyll b (27.6–9.8%) contents due to salt stress. There was also found significant increase in total chlorophyll contents (73.8–92.8%) with *Bacillus pseudomycoides* inoculum, while there was decrease (28.4–11.8%) with NaCl

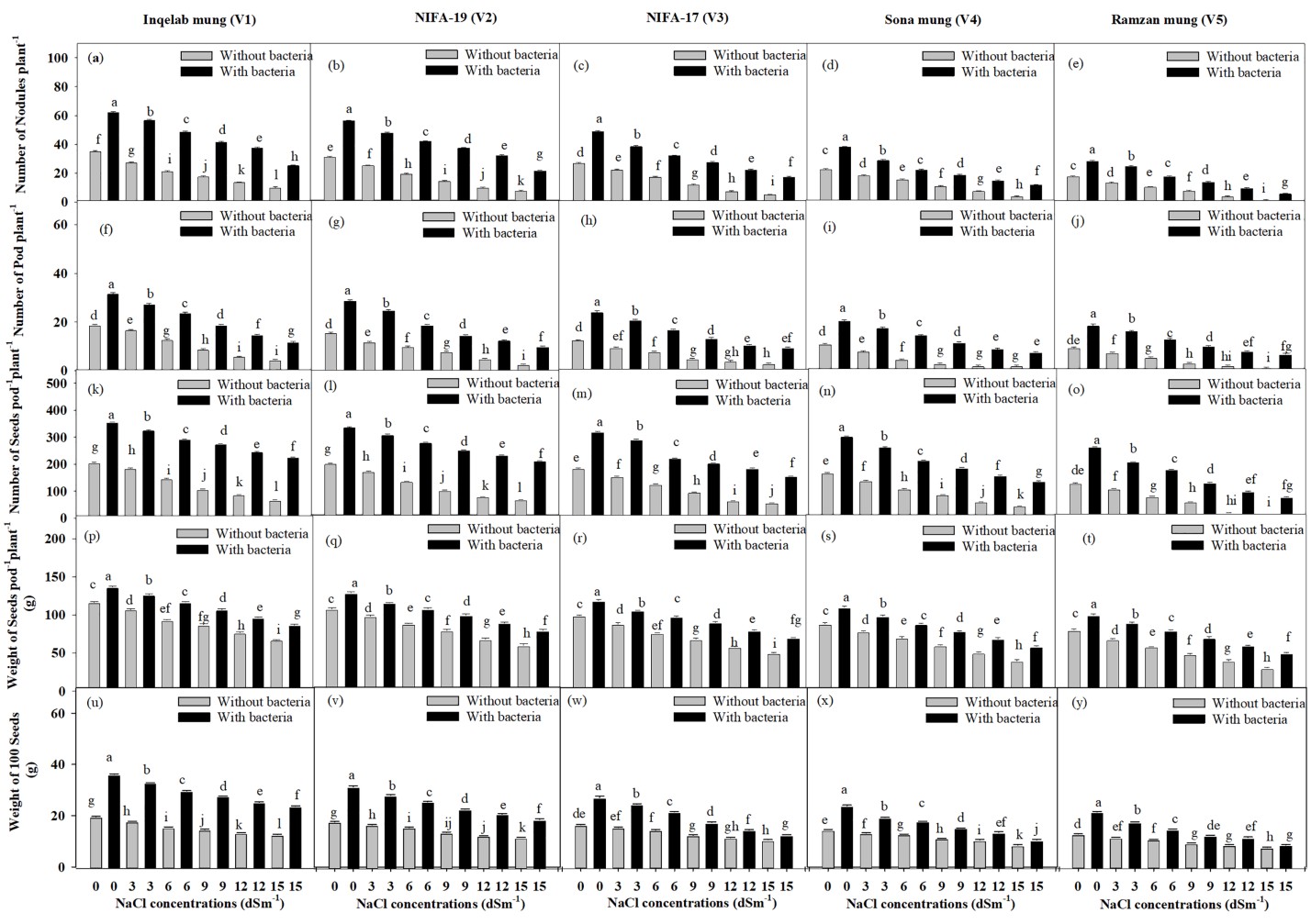

**Figure 3 Mean root nodules number plant$^{-1}$ (A–E), pods number plant$^{-1}$(F–J), seeds pod$^{-1}$plant$^{-1}$ (K–O), seeds weight pod$^{-1}$plant$^{-1}$ (P–T) and 100 seeds weight (U–Y) across five mung bean cultivars.** Bars indicate mean (±) standard error and different letters represent significant differences. Grey bars indicate soil with NaCl treatment, while black bars indicate NaCl treatment combined with *B. pseudomycoides* inoculum.

3–15 dSm$^{-1}$ across the cultivars, from Inqelab mung to Ramzan mung. Chlorophyll a contents were higher than chlorophyll b contents under all treatments in all the cultivars.

## Relationship among the variables

There was a significantly inverse relationship between plant biomass (fresh R$^2$ = 0.97–0.98 and dry-R$^2$ = 0.99–0.80)) and salt concentration across all the five cultivars. The inverse relationship was also observed among the number of root nodules (R$^2$ = 0.98–0.99), pods per plant (R$^2$ = 0.97–0.96), seeds per plant (R$^2$ = 0.98–0.86) and weight of seeds (R$^2$ = 0.98 −0.96) against salt concentration (Fig. 8).

## DISCUSSION

In this study, five cultivars of *V.radiata* were screened for their NaCl stress tolerance and the salt tolerant bacterial strain, *Bacillus pseudomycoides* was used to test whether it can mitigate the applied stress of NaCl. The selected five cultivars showed significant

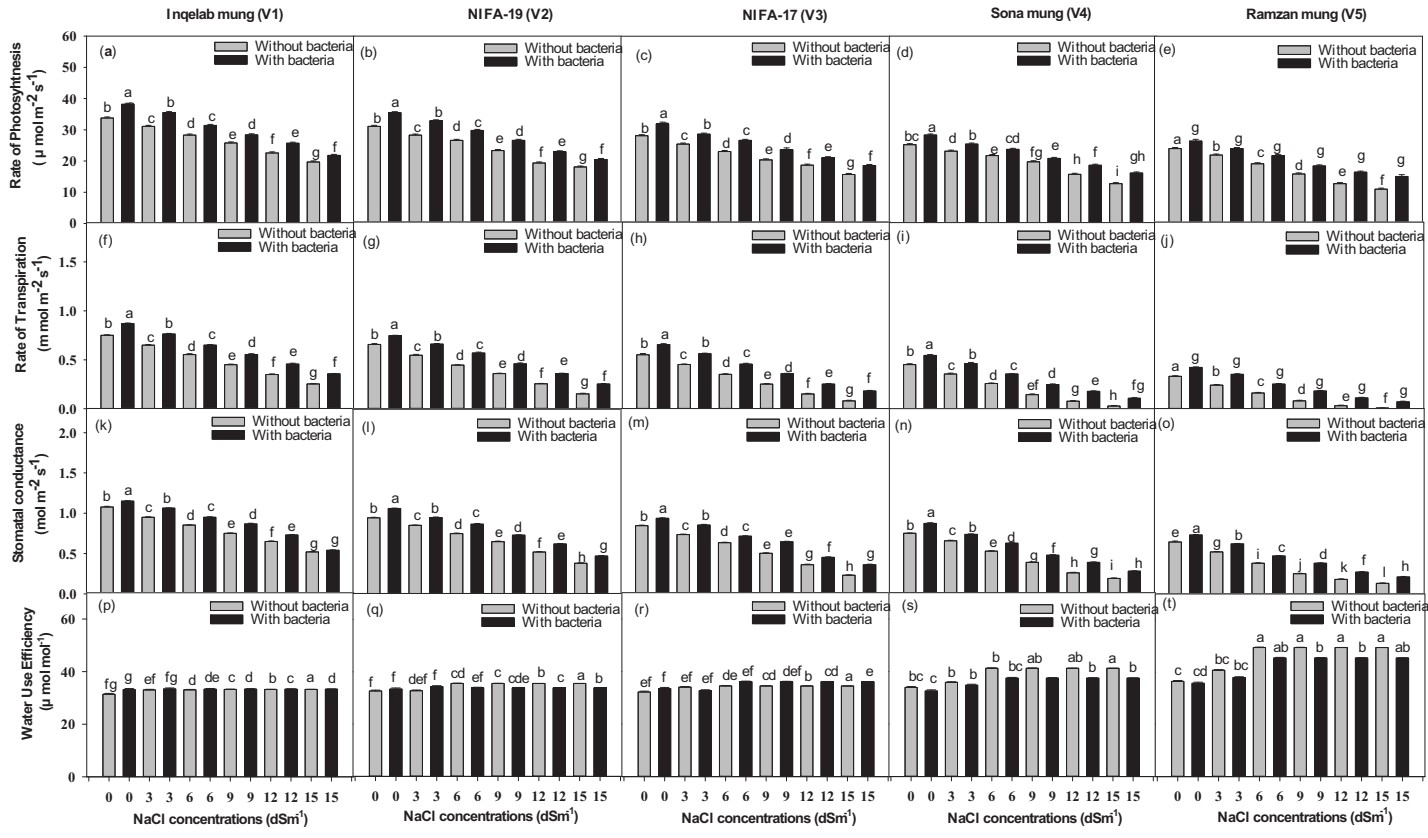

**Figure 4 The mean photosynthetic rate (A–E), transpiration rate (F–J), stomatal conductance (K–O) and water use efficiency (P–T) across the studied five mung bean cultivars.** Bars indicate mean (±) standard error and different letters mean significant differences. Grey bars indicate soil with NaCl treatment while black bars indicate NaCl treatment combined with *B. pseudomycoides*.

differences in terms of their tolerance towards NaCl stress with identified bacterial strain *i.e.*, *Bacillus pseudomycoides*. The application of *Bacillus pseudomycoides* together with the salt stress significantly enhanced the plants' growth and yield. This salt stress screening of the cultivars and its alleviation through *Bacillus pseudomycoides* has provided novel information for cultivating *V. radiata* cultivars in salt affected areas. Among the five studied cultivars, Inqelab mung showed more resistance to salt application than the other four cultivars. This could be due to the more nitrogen fixation ability and more uptakes of minerals by Inqelab mung, as it had larger number of root nodules, root hairs, balanced phytohormones, and production of reactive oxygen species as well as osmotic adjustment (*Mansoor et al., 2017*). *Rehman et al. (2019b)* also reported the interaction of rhizobium strains with mung bean cultivars "Inqelab mung" that showed bacterial strains elevated yield and physiological parameters as it increased nutrient absorption, systematic resistance against pathogens, phosphorus solubilization as well as production of phytohormones. *Khattak et al. (2020)* suggested NIFA-17 mung as a commercial crop for some areas in Northern Pakistan, because it provided more yield in terms of seed production, seed size and number of pods than Ramzan mung. *Khattak, Saeed & Ibrar (2019)* found NIFA-19 as a new commercial variety because it gave more yields, *i.e.*, more

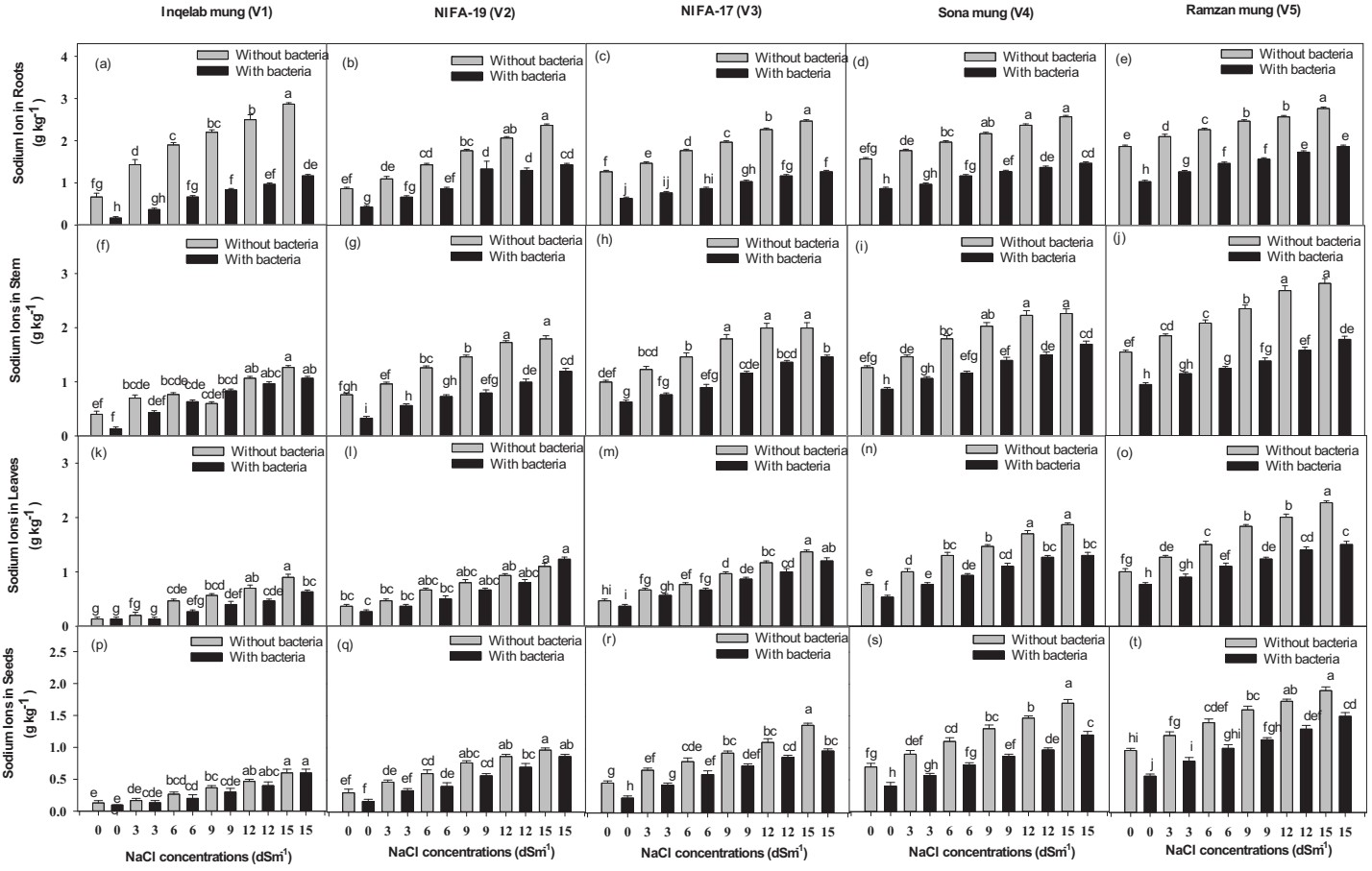

**Figure 5 Sodium ion uptake in root (A–E), stem (F–J), leaves (K–O) and seeds (P–T) across the five mung bean cultivars.** Bars indicate mean (±) standard error and different letters represent significant differences. Grey bars indicate soil with NaCl treatment and black bars indicate NaCl treatment combined with *B. pseudomycoides* inoculum.

seed size, number and production. Ramzan mung was also evaluated as a hybridized cultivar with short height, stiff stem, earlier maturity, and moderate grain with no resistance against diseases (*Khattak et al., 2006*).

Our findings showed that salt concentrations, *Bacillus pseudomycoides* inoculum and selected cultivar could increase biomass and yield. This shows that all these three factors could significantly enhance the yield among the mung bean plants. In our study, salt stress decreased biomass and yield of mung bean cultivars, while inoculum of *Bacillus pseudomycoides* improved them. This is in line with *Malook et al. (2020)*. A substantial imbalance in osmotic and water relation is caused by salinity stress, which declines plant growth. Salinity stress can be alleviated by using plant growth promoting bacteria, according to *Negacz et al. (2022)*. Inoculation of plant growth-promoting bacteria, such as *Acinetobacter bereziniae, Enterobacter ludwigii* and *Alcaligenes faecalis* ameliorated the harmful effect of saline stress in *Pisum sativum*. It also enhanced biomass and nutrient uptake efficiency. Our findings support the report of *Sapre, Gontia & Tiwari (2022)*, showing that biomass and yield of mung bean cultivars decreased with salt stress. This is because Na$^+$ ion level increased which induced oxidative stress, harmed ionic balance,
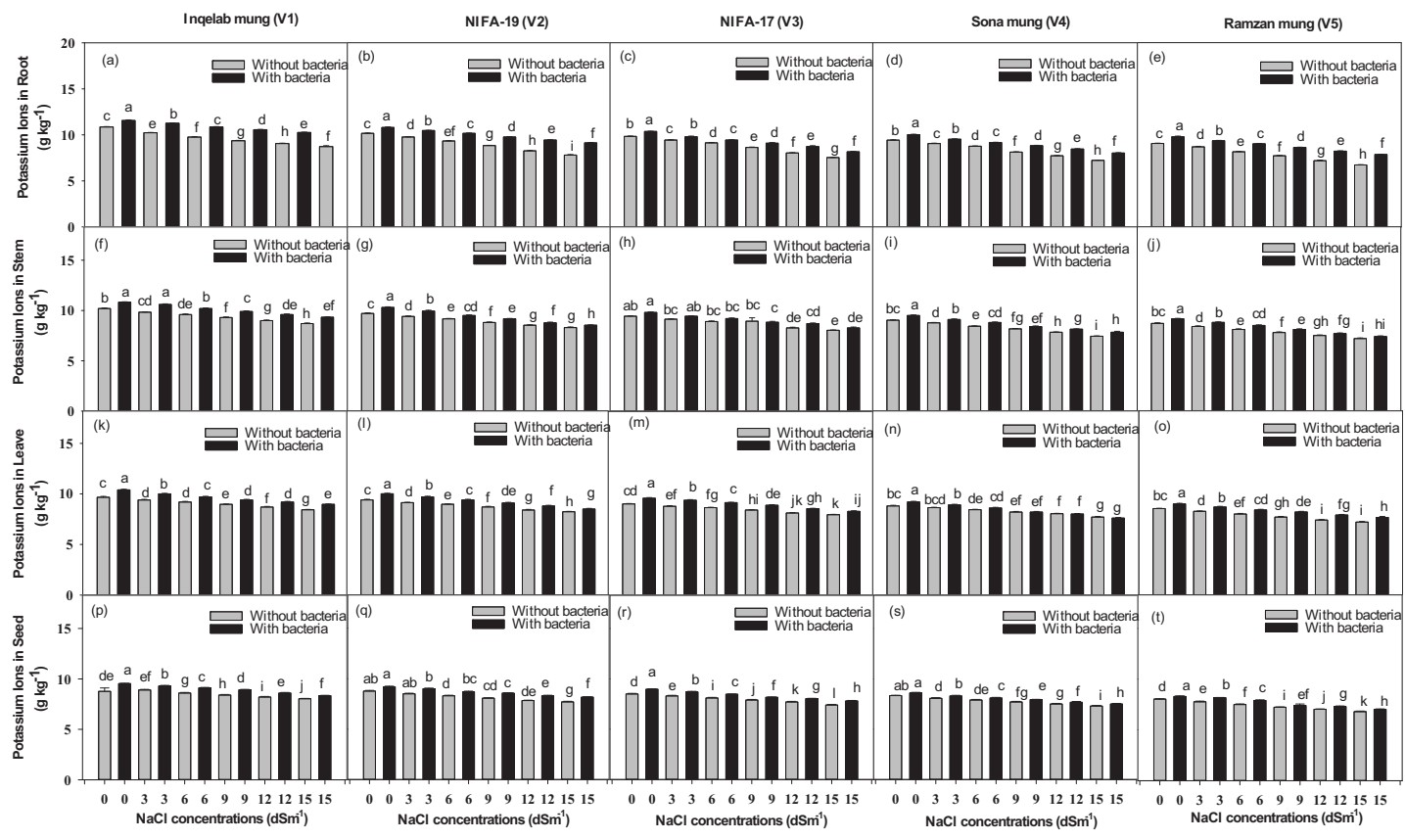

**Figure 6** Potassium ion uptake in root (A–E), stem (F–J), leaves (K–O) and seeds (P–T), across the five mung bean cultivars. Bars indicate mean (±) standard error and different letters mean significant differences. Grey bars indicate soil with NaCl treatment while black bars indicate NaCl treatment combined with *B. pseudomycoides* inoculum.

membrane integrity and photosynthetic efficiency. The inoculation of *B. pseudomycoides* improved biomass and yield significantly, as reported by *Katsenios et al. (2022)*, who treated sweet corn with different types of plant growth promoting bacteria. There was increased growth and yield in the following order: B. mojavensis, 16%; B. subtilis, 13.8%; B. pumilus, 11.8% and B. pseudomycoides, 9.8% compared to control. *Li et al. (2022)* also characterized Bacillus pseudomycoides as a heavy metal (Cu) resistant bacterium. Salt stress negatively affects root structure. *Youssef et al. (2023)* also found that NaCl stress had negative effect on the morphological properties of French beans. It caused enzymatic and metabolical degradation, physiological rate alteration, and imbalance in osmotic potential which destroyed the root structure. Bacillus sp. also worked symbiotically to improve V. radiata growth and yield that stabilized metabolism and osmotic balance (*Ahmed et al., 2020*).

In our findings, root nodules also suffered from salt stress, as confirmed by *Dogra et al. (2013)*. In their study, number of root nodules decreased in chickpeas under salt stress, while its effect was ameliorated by *Mesorhizobium cicero* strain which improved root nodules, due to its nitrogen fixation ability. *Hnini, Taha & Aurag (2022)* also found that rhizobacteria in *Vachellia tortilis* and the co-inoculation of *Enteribacter hormaechei*,

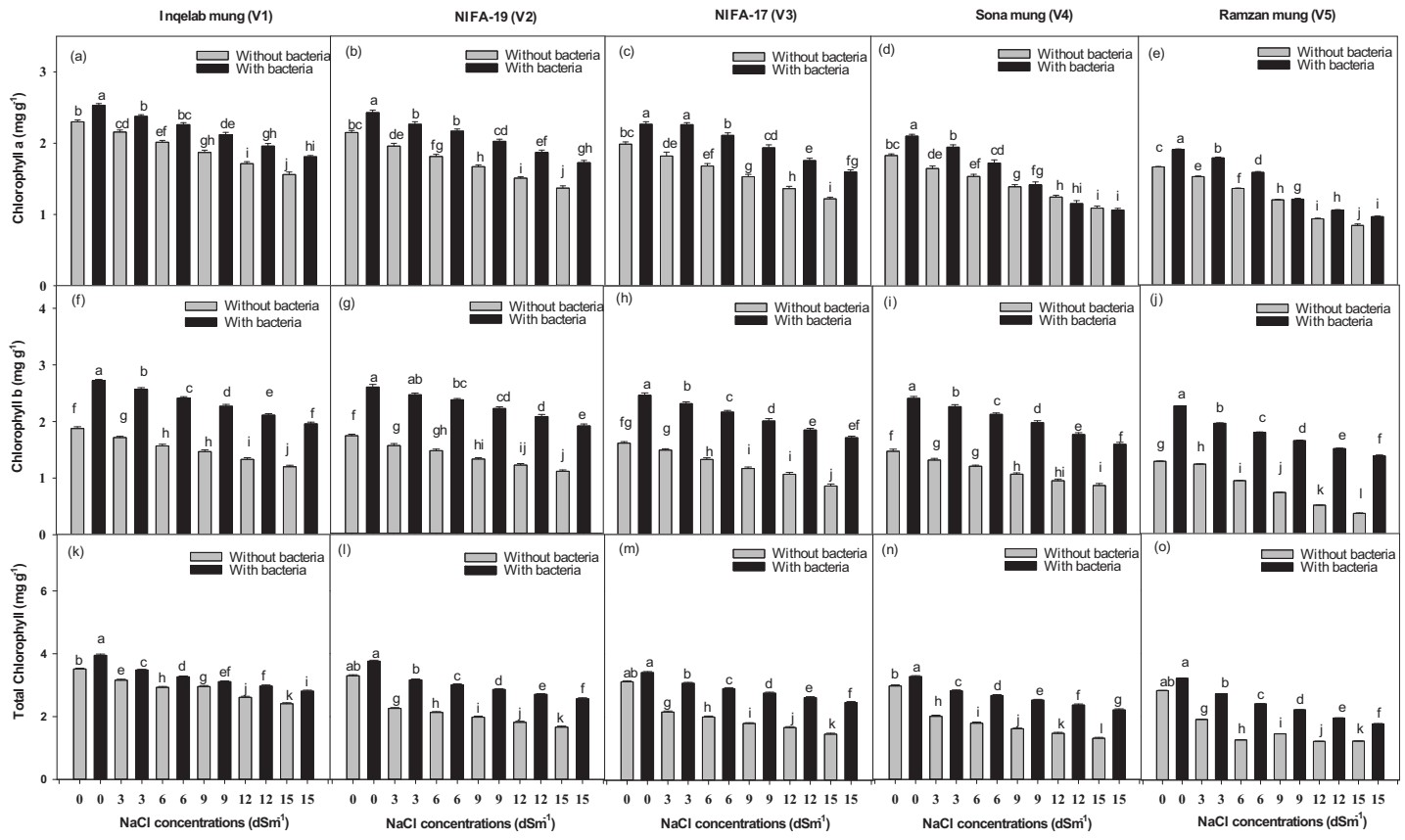

**Figure 7 Chlorophyll a (A–E), chlorophyll b (F–J) and total chlorophyll contents (K–O), across the five mung bean cultivars.** Bars indicate mean (±) standard error and different letter (s) represent significant differences. Grey bars indicate soil with NaCl treatment while black bars indicate NaCl treatment combined with *B. pseudomycoides* inoculum.

*Peudomonas moraviensis* bacteria in *Sulla flexuosa* L. (*Hamane et al., 2023*) developed more root nodules enhancing root colonizing potential and counteracting nutrient imbalance caused by salt stress.

Our results are also supported by *Medeiros & Bettiol (2021)*, who investigated the inoculation of tomatoes with *Bacillus* sp. It was revealed that the tomato plants treated with *Bacillus* sp had more yield than that of the plants under stress. The factors responsible for it could include increased phytohormones, more ROS scavenging ability as well uptake of minerals and reduced osmotic stress. *Bacillus pseudomycoides* can activate different enzymes, such as lipase, pectinase and cellulase that play a major role in minimizing stress and increasing plant yield (*Knezevic et al., 2021*). Under salt stress, chlorophyllase can break down different photosynthetic pigments, chlorophyll and other enzymes as well as overproduction of reactive oxygen species. The chlorophyll contents, rate of photosynthesis, transpiration and stomatal conductance were negatively affected by salt stress. Our findings are in line with *AbdelMotlb et al. (2023)*, who reported that salt stress reduced chlorophyll contents and physiological properties in green gram. The bacterial inoculum enhanced these parameters by increasing electron transport and photosynthetic activity. In our study, *B. pseudomycoides* also affected water use efficiency of mung bean

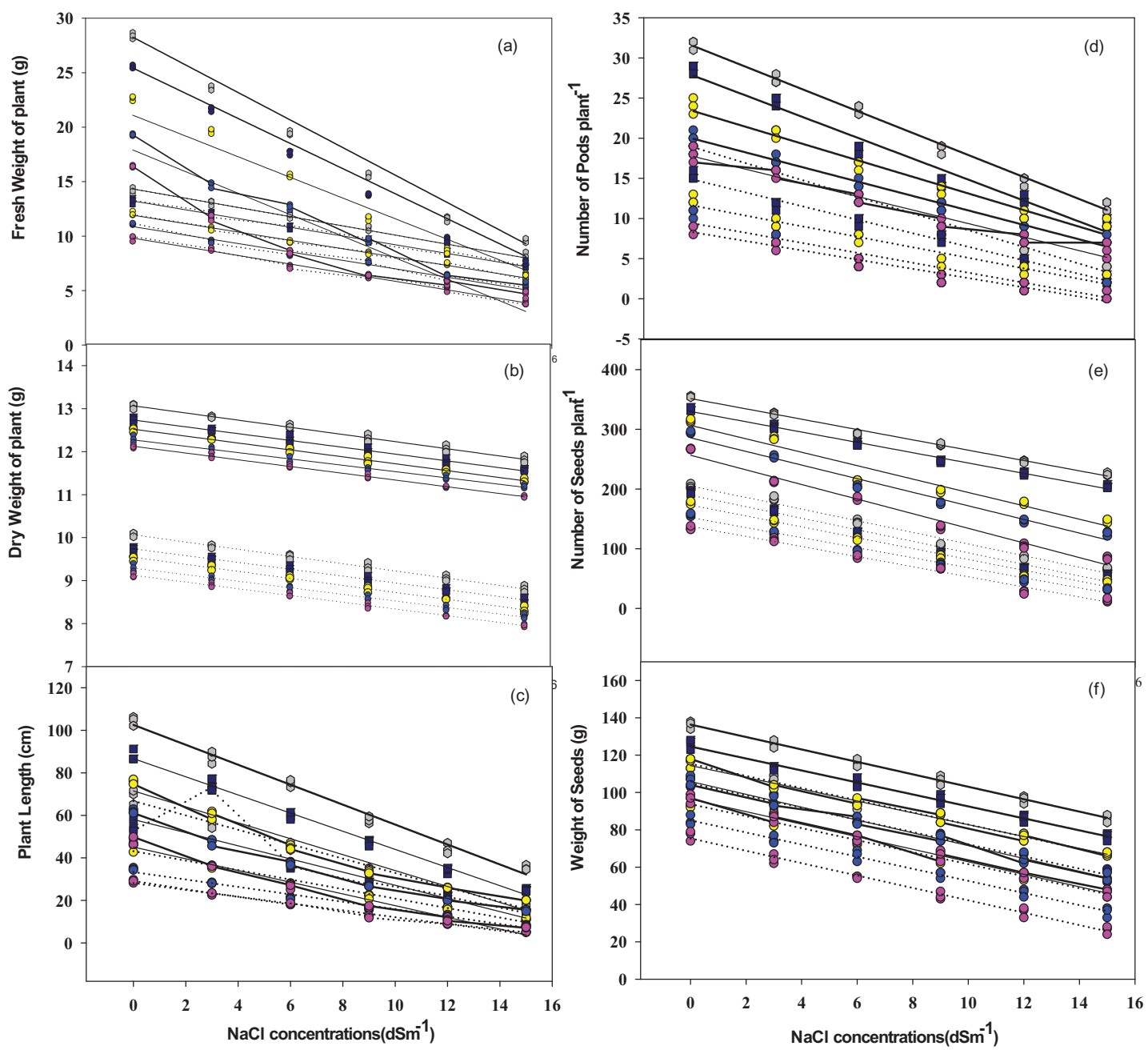

**Figure 8 Relationship of NaCl concentrations with plant fresh weight, dry weight, plant length (A–C), pods number plant[−1], seeds number plant[−1], and seeds weight (D–F) in mung bean cultivars.** Dotted lines indicate plant without bacteria inoculum, while solid lines represent plants with bacteria inoculum. Grey dots = V1, Dark blue dots = V2, Yellow dots = V3, Blue dots = V4, Purple dots = V5.

cultivars against salt stress. This is in line with the findings of *Obadi et al. (2023)* on tomatoes.

The ion uptake results showed that plants under salt stress had more sodium and less potassium uptake in their roots compared to their stems, leaves, and seeds, while those

plants treated with *Bacillus pseudomycoides* inoculum had improved potassium uptake. There was more influx of sodium ions in Ramzan mung, while Inqelab mung had less influx of sodium ions, among all the cultivars. Salt stress influences more sodium ion uptake in plant cells, which initiate osmotic stress, damaged plant tissue, disrupt cellular processes and membrane integrity. *Leontidou et al. (2020)* found similar result in tomatoes under salt stress. Higher levels of salt can lead to the accumulation of more sodium ions in plant cells, which requires more energy to maintain. Such energy which could be used for plant growth and development is wasted in combating stress. In the presence of excess salt, $Na^+$ influx depolarizes the root plasma membrane which activates the outward rectifying potassium channels in the guard cells. This activation provides a pathway for the diffusion of $Na^+$ into the cells, concomitantly decreasing cytosolic $K^+$ and increasing $Na^+$ content. Soil-inhabitant bacteria mitigate the uptake of sodium ions by forming an ion export system and preventing cell damage. *Ahmad et al. (2011)* demonstrated that inoculation of *Rhizobium* sp. and *Pseudomonas* sp. activated ion exclusion mechanisms that removed extra $Na^+$ from roots, which prevented seeds and leaf from absorbing salt. In the current study, *Bacillus pseudomycoides* played a major role in helping to overcome salt stress and enhance mung growth and yield. Also, *ElSharawy, Eid & Ebrahiem (2023)* found *B. pseudomycoides* to be a promising bio-control agent against *Alternaria* early blight in tomatoes.

Our results showed that *Bacillus psedomycoides* inoculum lowered $Na^+/K^+$ ratio, which boosted plant nodulation, yield and physiological attributes against applied stress. Higher $Na^+$ accumulation also resulted in a higher $Na^+/K^+$ ratio, which disrupted ion homeostasis. Similar reports were presented by *Zhang et al. (2018b)* and *Jiang et al. (2019)* that salt stress in rice, cucumber and *Cicer arietinum* increased $Na^+/K^+$ ratio by inducing oxidative stress, reducing antioxidant system and disrupting cellular functions; while inoculation of *Rhizobium* sp. proved to be significant against salt stress. These results are in agreement with previous findings of *Mushtaq et al. (2021)*. It is considered that bacterial inoculum supports different mechanisms, such as intra-cellular $Na^+$ sequestration, cyctolic potassium retention, osmotic adjustment though synthesis of compatible solutes that enhances crop growth and yield even under stress.

## CONCLUSION

The study concludes that five cultivars of mung bean (*Vigna radiata* L) had decreased growth and yield under NaCl stress. However, the combined effect of NaCl stress and *Bacillus pseudomycoides* inoculum significantly improved their growth, yield, and physiological properties. Of all the different plant parts, roots stored maximum amount of $Na^+$, while seeds accumulated minimum amount across all the cultivars (from Inqelab mung to Ramzan mung) under salt treatments. All of the parameters were found to decrease with increased NaCl stress (3–15 dSm$^{-1}$). Among all the cultivars, Inqelab mung showed the highest resistance to stress, while Ramzan mung demonstrated the lowest resistance.

### Funding

This work was supported by the Government College University Lahore, Pakistan (348/ORIC/23). The funders had no role in study design, data collection and analysis, decision to publish, or preparation of the manuscript.

### Grant Disclosures

The following grant information was disclosed by the authors:
Government College University Lahore, Pakistan: 348/ORIC/23.

### Competing Interests

The authors declare that they have no competing interests.

### Author Contributions

- Bushra Bilal conceived and designed the experiments, performed the experiments, analyzed the data, prepared figures and/or tables, authored or reviewed drafts of the article, and approved the final draft.
- Zafar Siddiq conceived and designed the experiments, analyzed the data, prepared figures and/or tables, authored or reviewed drafts of the article, and approved the final draft.
- Tehreema Iftikhar analyzed the data, authored or reviewed drafts of the article, and approved the final draft.
- Muhammad Umar Hayyat conceived and designed the experiments, performed the experiments, analyzed the data, prepared figures and/or tables, authored or reviewed drafts of the article, and approved the final draft.

### Data Availability

   The raw data are available in the Supplemental File.

### Supplemental Information

Supplemental information for this article can be found online at http://dx.doi.org/10.7717/peerj.17465#supplemental-information.

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
