# Peer review of "Mitigating NaCl stress in Vigna radiata L. cultivars using Bacillus pseudomycoides"

_PeerJ, doi:10.7717/peerj.17465_

## Round 0.1 · original submission · Major Revisions

· Academic Editor

Major Revisions

Authors are advised to revise the manuscript as per the suggestions of the reviewers. Therefore, I recommend that the manuscript be considered only after the corrections employed by the authors.

**Language Note:** The review process has identified that the English language must be improved. PeerJ can provide language editing services - please contact us at [email protected] for pricing (be sure to provide your manuscript number and title). Alternatively, you should make your own arrangements to improve the language quality and provide details in your response letter. – PeerJ Staff

Reviewer 1 ·

Basic reporting

The manuscript lacks the professional and unambiguous language required for scientific reporting. The text needs substantial revisions to enhance clarity, coherence, and adherence to academic language standards. The writing style should be refined to meet the expected standards of the journal.
The manuscript provides context but lacks depth in linking the research to existing literature. While some references are included, the background information does not sufficiently highlight the gap in knowledge that this research aims to fill.
The figures provided lack adequate description and labeling, affecting their overall quality. Additionally, the raw data supporting the findings have not been supplied as per the journal's policy. The authors should include high-quality, relevant figures with detailed descriptions and provide raw data to support their conclusions.

Experimental design

The research question is well defined and relevant. However, more explicit connection between the research question and the gap in the existing literature should be established.

Validity of the findings

The manuscript presents findings that are supported by robust data and statistical analyses, contributing to the overall validity of the study. The data presented are consistent with the research objectives and are controlled appropriately, enhancing the reliability of the conclusions drawn.

Additional comments

Language Enhancement: The authors should revise the manuscript to ensure professional, clear, and precise language throughout the document.
Literature Review Improvement: Strengthen the background by incorporating more relevant literature and explicitly stating how this research addresses the identified gap in knowledge.
Figure Quality and Raw Data Submission: Improve figure quality, ensure proper labeling and description, and provide the raw data to comply with the journal's policy.
Methodological Detail: Enhance the description of methods to allow for effective replication by other researchers.

·

Basic reporting

The manuscript is well written and has good professional english except some minor spelling mistakes at few places. Some of the references cited are irrelevant and they are mentioned in the original file as annotations. The introduction is well synthesized and possess significant background information to justify the study. however, the objectives of the study should be revised. suggestions are given in annotation.

Experimental design

The research questions are sufficient but they need to be revised thoroughly. I have suggested few points in annotation. The experimental design needs revision as salt concentration against each treatment are not labelled. moreover, the statistical design if included in the text can make this section more robust.

Validity of the findings

The data and results are supported with raw data therefore no ambiguity is observed. Since all the data is relevant to experimental procedures conducted in the research therefore validity can not be questioned.

Additional comments

Well synthesized manuscript but needs some revisions

---

## Round 0.2 · accepted · Accept

· Academic Editor

Accept

Authors have revised the manuscript as per suggestions of both reviewers. Therefore, I recommend that manuscript can be accepted for publication in PeerJ.

Reviewer 1 ·

Basic reporting

It is ok.

Experimental design

Experimental design is appropriate and sufficient detail is provided.

Validity of the findings

Corrections have been made by the authors in revised manuscript.

Additional comments

Authors have revised the manuscript as per reviewer's comments. Therefore, I recommend acceptance of this manuscript.